# The Use of Robotic-Assisted Bronchoscopy in the Diagnostic Evaluation of Peripheral Pulmonary Lesions: A Paradigm Shift

**DOI:** 10.3390/diagnostics13061049

**Published:** 2023-03-09

**Authors:** Hiba Hammad Altaq, Miloni Parmar, Talal Syed Hussain, Daouk J. Salim, Fawad A. Chaudry

**Affiliations:** 1Department of Pulmonary & Critical Care, University of Oklahoma, Oklahoma City, OK 73019, USA; 2Department of Internal Medicine, University of Oklahoma, Oklahoma City, OK 73019, USA

**Keywords:** peripheral lung lesions, outcomes, safety

## Abstract

Despite recent developments, evaluation of peripheral pulmonary lesions (PPL) remains clinically challenging, and the diagnostic yield of many image-guided and bronchoscopy methods is still poor. Furthermore, complications from such procedures, such as pneumothorax and airway hemorrhage, are a major concern. Recently launched robotic-assisted bronchoscopy (RAB) platforms are still in the early exploration stage and may provide another tool for achieving PPL evaluation. We present our experience here as a retrospective cohort study describing the 12-month diagnostic yield with the shape-sensing Ion™ platform for minimally invasive peripheral lung biopsy. The study describes forty-two patients undergoing shape sensing robotic-assisted bronchoscopy (ssRAB) at our institute. The early performance trend reveals a lesion localization of 100% and an overall 12-month diagnostic yield of 88.10%. The diagnostic yield for lesions less than 20 mm was 76% and for lesions greater than 20 mm was 100%. We also report our complication profile; we noted no pneumothoraces, excessive bleeding, or post-operative complications. In comparison to traditional bronchoscopy and image-guided modalities, our experience shows that ssRAB can be utilized successfully to travel to extremely small peripheral lesions with a higher diagnostic yield and better safety profile.

## 1. Introduction 

In the last decade, the detection of lung nodules has dramatically increased due to the adoption of lung cancer screening programs and the advances in computed tomography (CT) imaging. Approximately 1.6 million new pulmonary nodules are expected to be detected annually in the USA [1]. Clinicians must evaluate the clinical and radiological risk factors of these nodules. Depending on their size, shape, location, and risk factors, we can identify nodules at high risk for malignancy, which will warrant further diagnostic studies, such as biopsies [2].

The National Lung Screening trial has shown that most of these nodules (~80%) are usually located in the periphery of the lung [3,4]. If malignancy is a concern, histological confirmation is required to establish a more accurate diagnosis. One of the best-known non-surgical options to determine the etiology of a suspicious lung nodule is a CT-guided biopsy. The number of passes and size of the nodule are two criteria that could affect its accuracy [5]. Transthoracic needle aspiration and biopsy had a sensitivity of 90% (CI: 88–91%) and a specificity of 97% (CI: 96–98%) in studies examining its usefulness in making the diagnosis of peripheral bronchogenic carcinoma [6]. While false positive rate is low (average of 1%), the false negative rate is significant (average of 22%) [7,8]. In the absence of a malignant result, a benign diagnosis is therefore comforting; nonetheless, a non-diagnostic result should not be used to rule out a malignant outcome [9].

That said, this advantage comes with a significantly higher risk of pneumothorax for CT-guided core biopsy. The pooled rate of pneumothorax has been reported to be 25.3%, the rate of pulmonary hemorrhage 18.0%, and hemoptysis 4.1% [10]. Pneumothorax and hemoptysis occurred in less than 4% of the cases with ssRAB [11]. 

Other diagnostic approaches, such as virtual bronchoscopy (VB), radial endobronchial ultrasound (r-EBUS), electromagnetic navigation bronchoscopy (ENB), and ultra-thin bronchoscopes, have been used to obtain tissue biopsies [12]. However, these other diagnostic approaches tend to have a particularly low yield for lesions <2 cm in the outer third of the lung [13,14]. 

Here, we present the development of a new platform, namely shape-sensing robotic-assisted bronchoscopy (ssRAB), which provides the ability to navigate successfully to peripheral lesions with a higher diagnostic yield and better safety profile as compared to conventional bronchoscopy and image-guided modalities [11]. It is also important to point out that, in the majority of cases, endobronchial ultrasound (EBUS) lymph node staging was carried out following biopsy during the same procedure in order to possibly decrease CT-body divergence (single anesthetic event). The goal was to optimize workflow for a thorough diagnosis of concerning nodules with full staging in one procedure [15].

The primary objective was to evaluate the real-world performance characteristics of ssRAB in diagnosing peripheral pulmonary nodules. The safety profile of shape-sensing robotic bronchoscopy was also assessed.

## 2. Materials and Methods

We present a retrospective cohort study of our experience with the shape-sensing Ion™ endoluminal platform for minimally invasive peripheral lung biopsy. The study period (from the first enrollment to the last follow-up) was from 7 April 2021, to 9 March 2022. Our primary focus was the performance metrics of ssRAB in terms of the 12-month diagnostic yield, sensitivity, specificity, positive predictive value, and negative predictive value. Complications and safety data were also evaluated and reported. 

We enrolled 42 participants based on the following inclusion criteria. Male or female patients that were enrolled had to be 18 years of age or older, qualify for elective bronchoscopy, and have recent CT evidence of one or more solid or semi-solid pulmonary nodules measuring less than 3.0 cm in the longest dimension and any lung section. Bronchoscopy was requested for the subjects being considered for enrollment in this study to diagnose malignancy or benign illness. The University of Oklahoma, IRB Office of Human Research Participant Protection, granted the trial IRB approval. Enrolled study participants supplied written informed consent, and the Health Insurance Portability and Accountability Act’s standards (HIPPA) for maintaining study subject confidentiality were followed.

Patients who fulfilled the requirements for inclusion had or did not have a suspicious bronchus sign, which was defined as the existence of a bronchus leading to or contained within the target nodule as seen on CT imaging carried out using a standardized methodology and a slice thickness of 0.75–1 mm, which was done within 30 days before the bronchoscopy procedure.

We excluded pregnant or nursing patients and those with any medical contraindication for bronchoscopy. All study procedures were performed with general anesthesia in a dedicated operating room.

### 2.1. Procedure

Using CT scans taken prior to the procedure, Ion’s PlanPoint^TM^ software designed navigation pathways to the target lesion. Ion’s^TM^ ultra-thin robotic catheter (Figure 1), which has a 2.0 mm working channel and a 3.5 mm outer diameter, was guided along the pre-determined path during bronchoscopy. Radial endobronchial ultrasonography (r-EBUS) was used to confirm the location of the lesion and transbronchial biopsies were taken using fluoroscopy (Figure 2 and Figure 3). Along with traditional biopsy tools, custom-designed flexible needles that can fit through the catheter even in twisted airways, and span between 21 and 23 G, were utilized to collect biopsies. In every instance, a rapid on-site assessment (ROSE) was used.

We began with a flexible bronchoscopy to inspect the airway and do an airway clearance under general anesthesia, which was managed by a trained anesthesiologist. Patients were adequately ventilated with tidal volumes between 6 and 8 mL/kg of patients’ ideal body weight and a PEEP of 10–12 mm Hg if hemodynamics allowed. These anesthesia parameters were chosen because of concerns about atelectasis, which can diminish the biopsy yield. 

Afterwards, the existing endotracheal tube was connected to the robotic bronchoscopy system. The robotic bronchoscopy system consists of a tower with a monitor and a unit with arms that can insert, retract, and articulate an outer sheath (with an outside diameter of 6.0 mm) and an inner scope (with an outer diameter of 4.4 mm and a working channel of 2.1 mm) [16]. A 3D segmented airway model was produced by the Ion Plan Point Planning Station using data from a recent preoperative CT scan, which facilitated navigating to and locating the lesion. Lesion localization was defined as the ability to get within 1 cm of the target lesion center, as used previously in the NAVIGATE trial [17]. The rEBUS probe was inserted into the system’s working channel and replaced with a biopsy tool if a reliable signal was detected (Figure 1).

We obtained measurements in all three axes, which included the distance to the nodule and the catheter orientation. If necessary, readjustment and other CT spins were performed before obtaining biopsy samples to confirm the catheter location. All tissue biopsies were run. Rapid on-site cytology evaluation was available for intraoperative feedback. We also performed standard mediastinal staging or other components of the bronchoscopy if needed for further diagnosis. 

Regarding the procedure and post-procedure complications, all patients had a chest X-ray post-operatively to evaluate for pneumothorax, and the Nashville bleeding score was used to report bleeding complications. The Nashville bleeding system [18] consisted of the following: Grade 1, where patients required less than 1 minute of suctioning or wedging of the bronchoscope resulting in spontaneous cessation of bleeding. Grade 2, suctioning for more than one minute or wedging of the bronchoscope for persistent bleeding, and use of cold saline or diluted vasoactive substances. Grade 3 is the use of selective intubation or a balloon blocker. Lastly, Grade 4 is persistent selective intubation for >20 min, packed red blood cells transfusion, or the need for bronchial arterial embolization. All 42 patients included in our study were grade 1 where no major intervention for bleeding was needed; patients were discharged home the same day post-procedure and had a 30-day clinic follow-up to assess for delayed postprocedural complications. 

### 2.2. Statistical Analysis

The primary endpoint of the study was the diagnostic yield of ssRAB sampling for each pulmonary lesion. Initial pathology results that were negative for malignancy or indeterminate were referred to as negative for malignancy. Patients with benign initial results and repeat sampling confirming benign (non-malignant) pathology or follow-up imaging showing no progression were considered true negatives at 12 months. Patients with indeterminate results and repeat sampling showing benign pathology or follow-up imaging showing a decrease in size were considered true negatives at 12 months. Cases that were not proven to be true negatives based on the above criteria were considered false negatives. 

After a tumor board review of pathology, results were determined to be positive if malignancy was diagnosed, and a definitive treatment plan was made. There were no false positive results, as all malignant cases were considered truly malignant and were treated as such. The 12-month yield was calculated as the rate of true positives and true negatives. Sensitivity, specificity, positive predictive value (PPV), and negative predictive value (NPV) were also calculated. Univariate analyses were done to determine whether there were any predictors of yield.

## 3. Results

A retrospective cohort study was conducted on patients who underwent robotic-assisted bronchoscopy between 7 April 2021 and 9 March 2022. Forty-two patients were included, consisting of 19 men (45.24%) and 23 women (54.76%) with a moderate-to-high probability of primary or secondary pulmonary malignancy. Most patients had a clinically significant smoking history, with 73.81% being current or former smokers at the time of the procedure. In total, 26/42 (61.9%) of patients had a documented history of chronic lung diseases, such as COPD, emphysema, or asthma. Furthermore, 19/42 (45.24%) patients had a previous history of malignancy, with 3/42 (7%) patients having a history of lung cancer. Baseline characteristics are reported in Table 1. 

### 3.1. Target Nodule Characteristics 

Most of these nodules were located in the right upper lobes with a median size of 12 mm (10–18 SD) in the largest dimension. Bronchus sign was present in 25/42 (59.52%) of patients. Most (37/42, 88.10%) of the nodules were solid (Table 1). 

### 3.2. Primary End Point: Diagnostic Yield 

The early performance trend reveals a lesion localization of 100% and an overall 12-month diagnostic yield of 88.1%. The diagnostic yield for lesions less than 20 mm was 76% (32/37) and for lesions greater than or equal to 20 mm was 100%. 

Our study’s diagnostic yield was 88.1%. The initial biopsy results for ten patients was negative for malignancy. Five patients were classified as benign or non-malignant. This includes one patient whose repeat biopsy of the lesion was a caseating granuloma; two patients’ initial biopsies were benign, and pathology results indicated a fungal etiology; one patient’s follow-up imaging showed a decrease in size, where imaging was repeated after 12 months but biopsy was not repeated; and lastly, one patient’s initial pathology was a necrotizing granuloma and follow up imaging at one year remained stable.

Five patients’ results were non-diagnostic. One patient had a repeat biopsy, and the findings were consistent, with chronic inflammatory cells but no follow-up imaging was obtained. Two patients’ repeat biopsies showed necrotic tissue/atypical cells, and on follow-up imaging obtained at <1-year nodule size was stable or unchanged. One patient underwent lobectomy with findings of pancreatic adenocarcinoma. Lastly, one patient had follow-up imaging for surveillance where the nodule increased in size, but the patient chose not to proceed with further diagnostic studies. These results are summarized in Figure 4 and Table 2. 

A univariate analysis was also performed to assess for predictors of 12-month yield, and no factor was found to be a statistically significant predictor except the number of passes (Table 3). As the number of passes to obtain adequate specimens increases by 1, the 12-month diagnostic yield decreases by 29.7%. The univariate analysis showed that the diagnostic yield was independent of the lobar lesion location, size, and lesion density; however, 88% of our lesions were solid in density (Table 3). 

### 3.3. Secondary End Point: Adverse Events 

There was no major bleeding and per the Nashville grading system, 100% of our cases were grade 1. No pneumothorax occurred in any patients immediately or 30 days after the procedure. There was also no evidence of airway trauma observed on insertion or retraction of the robotic bronchoscope. Lastly, all patients completed the procedure, remained hemodynamically stable, and were discharged home the same day post-procedure.

## 4. Discussion 

Lung cancer is the leading cause of cancer-related deaths annually in the United States, which might be related to patients being diagnosed at advanced stages [19]. The National Lung Screening Trial showed that low-dose CT chest screening in high-risk individuals increased earlier-stage detection [1,20]. With an increasing number of chest CTs performed for cancer screening, the number of incidental pulmonary nodules is also growing, with one study estimating it to be ~1.57 million annually [21]. Subsequently, the detection of peripheral pulmonary nodules (PPN) will also increase. The radiographic features alone are not reliable for differentiating between a benign and malignant nodule, and biopsy is often indicated in most cases [22]. 

PPNs are difficult to biopsy using a conventional flexible bronchoscope. In the past decade, a multitude of techniques have been developed, such as virtual bronchoscopy (VB), fluoroscopic guided bronchoscopy, r-EBUS, TTNB, and ultrathin bronchoscopes. TTNB has the highest diagnostic yield of 90%, but it also has a higher rate of complications, such as pneumothorax, limiting its utility [3,22]. Another technique is ENB, where the diagnostic yield was 73%, which was noted in a large, multicenter, prospective study, where results were published in the NAVIGATIONAL trial in 2019 [17]. 

Shape-sensing robotic-assisted bronchoscopy (ssRAB) is a novel technology developed to improve the diagnostic sensitivity of PPN biopsies with fewer procedural complications. Ion robotic endoluminal system used shape-sensing technology instead of electromagnetic navigation, which was FDA-approved in 2019 [12]. 

Limited studies have been published looking at the feasibility of ssRAB in a real-world setting, and ssRAB approaches remain in the early phases of exploration. Our data confirmed that the use of ssRAB success was similar to a recently reported study where the navigation rate was 98.7% with an overall diagnostic yield of 81.7% [11]. 

In this real-world, retrospective study, index test results were considered positive after yield was defined as a pathologic result that prompted a definitive treatment plan based on the Tumor Board review of the clinical case. Our definition of ssRAB success is meaningful for clinicians as it consists of diagnostic material on final pathology or r-EBUS image confirmation. 

Diagnostic yield via flexible bronchoscopy has been reported to be as low as 14% for peripheral nodules under 2 cm in lesions located in the outer third of the lung, making the size and location of the lesion a determinant for diagnosis [23]. 

In our study, the univariate analysis showed that the diagnostic yield was independent of the lobar lesion location, size, number of passes, and lesion density. 

These findings are similar to the previous experience of Chadda et al. [24]. However, 88% of our lesions were solid in density. The ability to use the robotic system to locate and confirm the lesion suggests that the current system successfully positions bronchoscopists close to targeted lesions, which is a critical step before performing biopsies. Finally, the safety profile was favorable, with a low risk for pneumothorax, airway damage, or any significant bleeding in patients. 

Some of the strengths of this study are that the yield was 88.1%, independent of lesion size and nodule location, and even though we did not use cone-beam CT (CBCT), our accuracy and yield remained high. In addition, although adverse events were not the primary endpoint, our assessment is that the procedural protocol is an additional strength of the study and may be helpful with future investigations to prevent post-op complications. Lastly, the diagnostic criteria were designed to protect against any potential ambiguity, and rapid on-site evaluation (ROSE) was required during the procedure to optimize the quality of tissue obtained. ROSE has become an important tool in the diagnosis and staging of malignant tumors of the lungs and mediastinum [25]. A preliminary result provides a valuable preliminary intraprocedural assessment of sample adequacy and improvement in diagnostic sensitivity [26].

R-EBUS was used in all the cases. There is a theoretical risk that false positive r-EBUS images are produced during bronchoscopy given passive atelectasis [27]. This likelihood increases with procedure time. In our cases, we performed robotic bronchoscopy prior to convex EBUS to decrease this risk.

The limitations of this study would be related primarily to the relatively single-arm non-randomized design, small sample size, single-center, retrospective study, and the 42 cases were performed by a single operator in a large university setting. We also believe that for other operators who do not perform this procedure often, there is a potential inability to generalize the results [16]. 

In addition, as highlighted above, not all negative cases were followed; if repeat imaging showed no progression or a repeat biopsy showed negative results, cases were true negative without the need to follow them for 12 months. 

Our research shows that, compared to traditional navigational bronchoscopy and image-guided modalities, ssRAB can successfully navigate extremely small peripheral lesions with a greater diagnostic yield and better safety profile. Our findings are in line with and are comparable to the scant information on ssRAB for PPL that is currently available globally. The precise diagnostic yield of ssRAB needs to be further characterized, and the uncommon possible complications need to be investigated in larger prospective, preferably randomized trials.

## Figures and Tables

**Figure 1 diagnostics-13-01049-f001:**
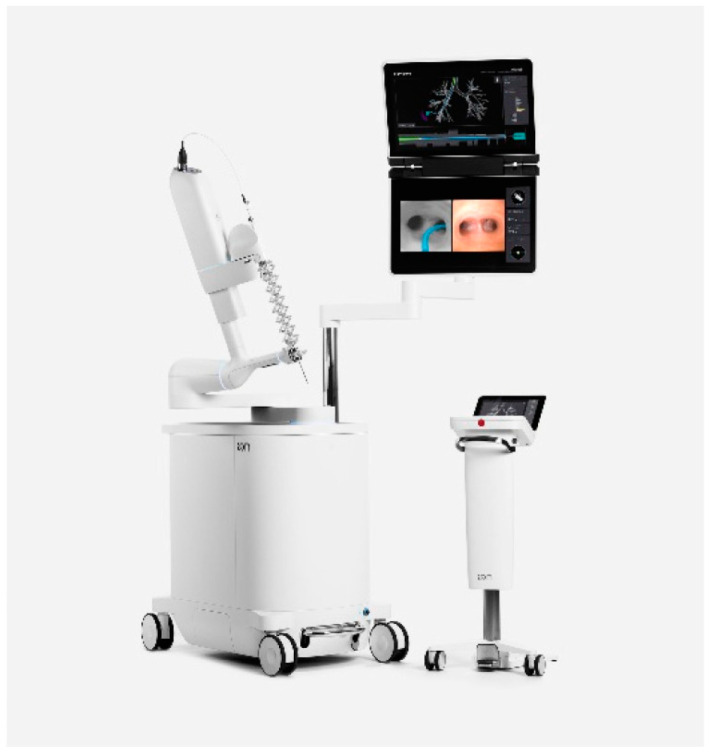
Ion™ Endoluminal System by Intuitive Surgical.

**Figure 2 diagnostics-13-01049-f002:**
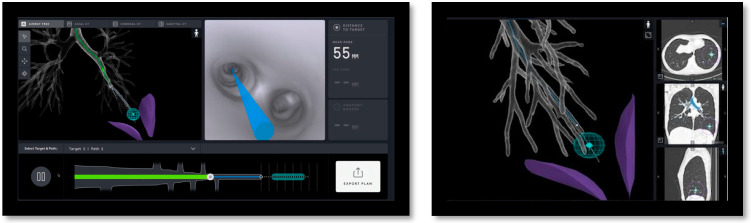
Procedural screenshots from a case. *Left* screen: Virtual airway view showing catheter within the airway tree. *Right* screen: Virtual target with target view.

**Figure 3 diagnostics-13-01049-f003:**
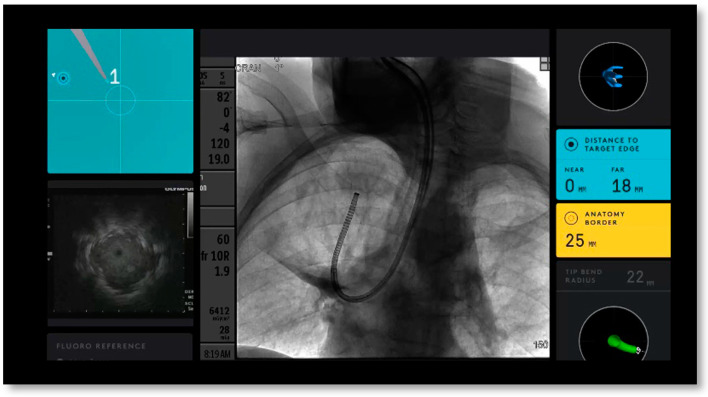
*Upper left* blue ball represents the target. *Bottom left* intergraded EBUS view. *Center* Fluoroscopy view of catheter with tool extension. *Right* Informational screen displaying distance to virtual target, anatomy borders, orientation guide, and catheter tip bend radius.

**Figure 4 diagnostics-13-01049-f004:**
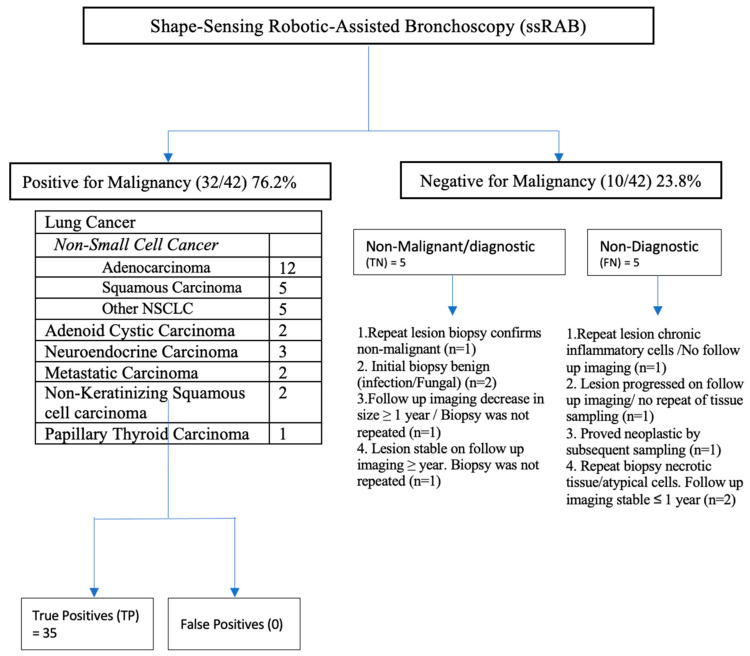
Algorithm describing diagnostic results to determine diagnostic outcomes in subjects undergoing robotic-assisted bronchoscopy lesion biopsy. Positive malignancy test results were considered positive pathologic results. For this 12-month follow-up study, pathology results that were negative for malignancy or indeterminate were referred to as negative for malignancy. These patients who had either repeat sampling confirming benign pathology or follow-up imaging showing no progression and were considered to be true negatives at 12 months. Cases that were negative and repeat sampling showed malignant pathology were considered to be false negatives at 12 months. One case was deferred in the primary analysis of the 12-month yield because of increased lesion size and no subsequent sampling because the patient refused.

**Table 1 diagnostics-13-01049-t001:** Demographics, lesion, and procedural characteristics.

Demographics	*N* = 42 Subjects
Female, No. (%)	23/42 (54.7)
Age, mean (SD), y *n* = 42	68.89 (11.8)
BMI, mean (SD), Kg/m^2^ *n* = 42	27.14(5.6)
Chronic lung disease (COPD/ILD/Asthma) No. (%)	26/42 (61.9)
History of cancer No. (%)	19/42 (45.2)
History of lung cancer No. (%)	3/42 (7)
Smoking history No. (%)	31/42 (73.8)
Lesion Properties	
Median Nodule size (mm) *n* = 42 (IQR)	12 (10–18)
FDG avid No. (%)	27/32 84.4%
Median SUV if FDG avid (mm) *n* = 27 (IQR)	4 (2.1–6.7)
Bronchus sign, No. (%)	25/42 (59.2)
Nodule location, No. (%)	
Right upper lobe	19/42 (45.2)
Left upper lobe	10 /42(23.8)
Right middle lobe	4/42 (9.5)
Right lower lobe	6/42 (14.3)
Left lower lobe	3/42 (7.1)
Lesion in central third of the lung	12/42 (28.6)
Lesion in middle third of the lung	20/42 (47.6)
Lesion in outer third of the lung	10/42 (23.8)
*Nodule Radiologic features No. (%)*	
Solid	37/42 (88)
Semi Solid	2/42 (4.8)
Ground glass	3/42 (7.1)
Spiculated lesion border	10 /42 (23.8)
Procedure characteristics	
General anesthesia	42 (100%)
Radial EBUS used during ENB	42 (100%)
Endobronchial ultrasound (EBUS-TBNA) No. (%)	23/40 (57.5)
*Bronchioalveolar lavage No. (%)*	
*+Malignancy*	4/42 (9.5%)
*+Infection Fungi*	2/42 (4.7)
*Negative (normal cellular analysis)*	36/42 (85%)
Fluoroscopy used during ENB	42 (100%)
ROSE used	42 (100%)
Median number of passes (mm) *n* = 35 (IQR)	10 (7–2)

Data are presented as % (n/N), median (interquartile range), or mean (standard deviation). EBUS, endobronchial ultrasound. ENB electromagnetic navigation bronchoscopy. EBUS-TBNA, endobronchial ultrasound-guided transbronchial fine needle aspiration. ROSE, rapid on-site evaluation.

**Table 2 diagnostics-13-01049-t002:** Twelve-month outcomes.

12-month yield	88.10
Sensitivity	86.49
Specificity	100.00
Positive predictive value	100.00
Negative predictive value	50.00

**Table 3 diagnostics-13-01049-t003:** Univariate analysis: Predictors of diagnostic yield.

Factor	OR	95% CI Lower Limit	95% CI Higher Limit	*p* Value
age	0.952	0.874	1.038	0.2646
Sex F vs. M	0.559	0.091	3.446	0.5307
BMI	1.017	0.867	1.192	0.8350
Diabetes	>999.999	<0.001	>999.999	0.9618
HTN	0.447	0.073	2.759	0.3861
Cardiovascular disease	1.000	0.160	6.255	1.0000
History of cancer	1.789	0.290	11.035	0.5307
Chronic lung disease	0.280	0.030	2.649	0.2669
Current or former smoking hx	0.520	0.054	5.021	0.5719
Py smoking hx	0.012	0.965	1.061	0.6271
EBUS done	0.633	0.102	3.938	0.6244
Multiple lesions samples	>999.999	<0.001	>999.999	0.9867
**Number of passes**	**0.703**	**0.497**	**0.995**	**0.0465**
spiculated	1.667	0.171	16.225	0.6600
Bronchogenic sign	0.250	0.026	2.361	0.2263
FDG avid	<0.001	<0.001	>999.999	0.9686
SUV	2.549	0.950	6.841	0.0632
size	1.137	0.908	1.425	0.2638
Location RUL	reference			
LLL	0.235	0.014	3.917	0.4210
LUL	1.059	0.084	13.329	0.4923
RLL	0.588	0.044	7.914	0.9494
RML	0.353	0.024	5.231	0.6578
Type solid	reference			
Ground glass	0.313	0.024	4.119	0.9657
Semi-solid	>999.999	<0.001	>999.999	0.9705
Centrality: outer	Reference			
Central	>999.999	<0.001	>999.999	0.9525
middle	1.000	0.150	6.671	0.9525

## Data Availability

The data that support the findings of this study are available from University of Oklahoma but restrictions apply to the availability of these data, which were used under license for the current study, and so are not publicly available. Data are however available from the authors upon reasonable request and with permission of University of Oklahoma.

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
