# Peer review of "The Use of Robotic-Assisted Bronchoscopy in the Diagnostic Evaluation of Peripheral Pulmonary Lesions: A Paradigm Shift"

_diagnostics, 2023, doi:10.3390/diagnostics13061049_

Round 1

Reviewer 1 Report

General comments:

This study evaluated the 12-month diagnostic yield of shape-sensing robotic-assisted bronchoscopy (ssRAB). Although this theme is one of the topics of pulmonary intervention field; however. I think this study has a lot of problems, especially in method section. This paper is difficult to evaluate because it sets its own diagnostic criteria that are difficult to interpret.

It needs to be revised with reference to a published paper on a similar topic.

Major comment 1:

The authors defined positive if a pathologic result prompted a definitive treatment plan based on the Tumor Board review of the clinical case. However, it is difficult to consider the criteria as appropriate due to ambiguity, because we cannot know what the pathology results were, whether the curative treatment was appropriate, and how the treatment plan was discussed in the Tumor Board review.

This criterion may result in an unjustifiably high diagnosis rate. The authors should change the definition of positive result to pathological diagnosis like the cited reference 11(Kalchiem-Dekel O, Connolly JG, Lin IH, et al. Shape-Sensing Robotic-Assisted Bronchoscopy in the Diagnosis of Pulmonary 328 Parenchymal Lesions. Chest. 2022;161(2):572-582. doi:10.1016/j.chest.2021.07.2169).

The criteria of true negative also should be reconsidered more rigorously like the cited reference above. The negative result of repeat biopsies does not mean true negative, and the study period is too short. At least one year follow up period for each nondiagnostic patient would be needed, unless the lesion shrinks or disappears.

Minor Comment 1:

Abstract. The abbreviation “ssRAB” is used without explanation in abstract.

Minor Comment 2:

Introduction. In line 46-47, does “Navigational biopsy” mean biopsy with ssRAB? The phrase "Navigational biopsy" seems ambiguous.

Minor Comment 3:

Introduction. In line 51-52, “higher complications risk” cannot be read from the cited literatures, therefore, the sentence is inappropriate.

Minor Comment 4:

Introduction. Some cited reference would be needed to support the sentence in line 53-56.

Minor Comment 5:

Method section. In line 71, What is “in this initial stage”? I could not find the “pre-determined criteria” The authors should explain clearly.

Minor Comment 6:

Method section. Why did you set a lower limit of 0.9 cm or more for the size of the target lesion instead of a requirement of 3 cm or less?

Minor Comment 7:

Method section. Explanations and references for Nashville bleeding score should be written in the Method section, not in the Result section.

Minor Comment 8:

Method section. It is difficult to understand the difference between Ion'sTM ultra-thin robotic catheter (2.0 mm working channel and a 3.5 mm outer diameter) and an inner scope (with an outer diameter of 4.4 mm with a working channel of 2.1 mm).  It would be easier to understand if you could add some figure or picture.

Minor Comment 9:

Method section. In line 137-139, the patient with negative pathology had increased lesion size on imaging with no repeat sampling after being followed for 12 months should be classified as a false-negative finding and nondiagnostic like the cited reference 11. And, the sensitivity analysis for the case is not needed.

Minor Comment 10:

Result section. The decimal point of the percentages should be aligned to one place.

Minor Comment 11:

Result section. In line 155-156, “(19)” is unnecessary.

Although a median size is shown with SD in text, it is shown with IQR in Table1. Which is correct?

Minor Comment 12:

Result section. Is the unit “(mm)” of “Median SUV if FDG avid” correct?

Minor Comment 13:

Result section. In Table 1, does “Endobronchial ultrasound (EBUS)” mean rEBUS or EBUS-TBNA?

Minor Comment 14:

Result section. In Table 1, does “Negative” of bronchoalveolar lavage means being not performed or being performed with no specific diagnosis?

Minor Comment 15:

Result section. In Table 1, What does “median number passes (mm)” mean?

Minor Comment 16:

Result section. Authors should add information about the number of biopsies and findings of rEBUS.

Minor Comment 17:

Result section. In line 163, the authors showed a lesion localization of 100%. What is the definition of the lesion localization? Please show the definition in Method section.

Minor Comment 18:

Result section. The authors should show a Table about the results of the univariate analysis.

Minor Comment 19:

Discussion section. The Abbreviations that are already use in main text should be used only with abbreviations thereafter.

Minor Comment 20:

Discussion section. In line 230, “{12]” has misdescription in parentheses

Minor Comment 21:

Discussion section. In line 232-234, the reference to comparisons of navigation rate and diagnosis rate with the cited papers seems inappropriate, because definition of navigation rate in this paper is unclear and difference of definition of diagnostic between this paper and the cited paper.

Author Response

Major comment 1:

The authors defined positive if a pathologic result prompted a definitive treatment plan based on the Tumor Board review of the clinical case. However, it is difficult to consider the criteria as appropriate due to ambiguity, because we cannot know what the pathology results were, whether the curative treatment was appropriate, and how the treatment plan was discussed in the Tumor Board review.

This criterion may result in an unjustifiably high diagnosis rate. The authors should change the definition of positive result to pathological diagnosis like the cited reference 11(Kalchiem-Dekel O, Connolly JG, Lin IH, et al. Shape-Sensing Robotic-Assisted Bronchoscopy in the Diagnosis of Pulmonary 328 Parenchymal Lesions. Chest. 2022;161(2):572-582. doi:10.1016/j.chest.2021.07.2169).

The criteria of true negative also should be reconsidered more rigorously like the cited reference above. The negative result of repeat biopsies does not mean true negative, and the study period is too short. At least one year follow up period for each nondiagnostic patient would be needed, unless the lesion shrinks or disappears.

 We have updated and changed our true negative results. It now brings us to a diagnostic yield of 88.10 %.

Our study diagnostic yield was 88.1%. Ten patients' initial biopsy results were negative for malignancy. Five patients were classified as benign or non-malignant. This includes one patient where repeat lesion biopsy was a caseating granuloma; two patients’ initial biopsy were benign, pathology results indicated fungal etiology; One patient follow up imaging decrease in size, imaging was repeated after 12 months but biopsy was not repeated; Lastly, one patient initial pathology was a necrotizing granuloma and follow up imaging remained stable at one year follow up. Five patient’s results were non diagnostic. One patient had a repeat biopsy, and the findings were consistent, with chronic inflammatory cells but no follow up imaging was done. Two patients repeat biopsy necrotic tissue/atypical cells, follow up imaging nodule size was stable or unchanged, but imaging was done less than one year < 1 year. One patient underwent lobectomy with findings of pancreatic adenocarcinoma. Lastly, one patient had follow-up imaging for surveillance where the nodule increased in size, but the patient has chosen not to proceed with further diagnostic studies.

Minor Comment 1:

Abstract. The abbreviation “ssRAB” is used without explanation in abstract.

Updated on manuscript

Minor Comment 2:

Introduction. In line 46-47, does “Navigational biopsy” mean biopsy with ssRAB? The phrase "Navigational biopsy" seems ambiguous.

Updated on manuscript

Minor Comment 3:

Introduction. In line 51-52, “higher complications risk” cannot be read from the cited literatures, therefore, the sentence is inappropriate.

Updated on manuscript

Minor Comment 4:

Introduction. Some cited reference would be needed to support the sentence in line 53-56.

Updated on manuscript

Minor Comment 5:

Method section. In line 71, What is “in this initial stage”? I could not find the “pre-determined criteria” The authors should explain clearly.

 Updated on manuscript

Minor Comment 6:

Method section. Why did you set a lower limit of 0.9 cm or more for the size of the target lesion instead of a requirement of 3 cm or less?

Updated on manuscript

Minor Comment 7:

Method section. Explanations and references for Nashville bleeding score should be written in the Method section, not in the Result section.

 Updated on manuscript

Minor Comment 8:

Method section. It is difficult to understand the difference between Ion'sTM ultra-thin robotic catheter (2.0 mm working channel and a 3.5 mm outer diameter) and an inner scope (with an outer diameter of 4.4 mm with a working channel of 2.1 mm).  It would be easier to understand if you could add some figure or picture.

Image added . Updated on manuscript

Minor Comment 9:

Method section. In line 137-139, the patient with negative pathology had increased lesion size on imaging with no repeat sampling after being followed for 12 months should be classified as a false-negative finding and nondiagnostic like the cited reference 11. And, the sensitivity analysis for the case is not needed.

Updated on manuscript

Minor Comment 10:

Result section. The decimal point of the percentages should be aligned to one place.

Updated on manuscript

Minor Comment 11:

Result section. In line 155-156, “(19)” is unnecessary. 

Although a median size is shown with SD in text, it is shown with IQR in Table1. Which is correct?

Updated on manuscript

Minor Comment 12:

Result section. Is the unit “(mm)” of “Median SUV if FDG avid” correct?

Correct. Updated on manuscript

Minor Comment 13:

Result section. In Table 1, does “Endobronchial ultrasound (EBUS)” mean rEBUS or EBUS-TBNA?

 Updated on manuscript

Minor Comment 14:

Result section. In Table 1, does “Negative” of bronchoalveolar lavage means being not performed or being performed with no specific diagnosis?

Updated on manuscript table

Minor Comment 15:

Result section. In Table 1, What does “median number passes (mm)” mean?

Updated on manuscript

Minor Comment 16:

Result section. Authors should add information about the number of biopsies and findings of rEBUS.

Updated on manuscript

Minor Comment 17:

Result section. In line 163, the authors showed a lesion localization of 100%. What is the definition of the lesion localization? Please show the definition in Method section.

 Updated on manuscript

Minor Comment 18:

Result section. The authors should show a Table about the results of the univariate analysis.

Table 3 . Updated on manuscript

Minor Comment 19:

Discussion section. The Abbreviations that are already use in main text should be used only with abbreviations thereafter.

 Updated on manuscript

Minor Comment 20:

Discussion section. In line 230, “{12]” has misdescription in parentheses

 Updated on manuscript

Minor Comment 21:

Discussion section. In line 232-234, the reference to comparisons of navigation rate and diagnosis rate with the cited papers seems inappropriate, because definition of navigation rate in this paper is unclear and difference of definition of diagnostic between this paper and the cited paper.

Updated on manuscript

Reviewer 2 Report

This is a valuable report that demonstrates the very high diagnostic performance and safety of ssRAB.

Here are some comments.

1. P1, L15. Please give the full spelling of ssRAB.

2. P6, Figure 3. Are Non-keratinizing Squamous cell carcinoma and Papillary Thyroid Cacinoma not included in metastatic tumors?

3. P4, L135-6. Lesions that had not increased for 12 months were considered true negatives, but it is possible, for example, for adenocarcinoma presenting with GGN, to have no change for a year. I wonder if any of these cases were included in the negative results this time.

4. Methods. Am I correct in assuming that all malignant diagnoses are tissue diagnoses and do not include cytology-only diagnoses?

5. Methods. Are all biopsy methods forceps biopsies? Are aspiration needle biopsies included?

Author Response

REVIEWER 1

P1, L15. Please give the full spelling of ssRAB. 

Updated on manuscript

  1. P6, Figure 4. Are Non-keratinizing Squamous cell carcinoma and Papillary Thyroid Cacinoma not included in metastatic tumors? Correct, they are not

  1. P4, L135-6. Lesions that had not increased for 12 months were considered true negatives, but it is possible, for example, for adenocarcinoma presenting with GGN, to have no change for a year. I wonder if any of these cases were included in the negative results this time. These lesions are now considered to be non-diagnostic findings if repat biopsies remained inconclusive or imaging nodule did not change in size.

  1. Methods. Am I correct in assuming that all malignant diagnoses are tissue diagnoses and do not include cytology-only diagnoses? Correct they are tissue diagnosis

  1. Methods. Are all biopsy methods forceps biopsies? Are aspiration needle biopsies included? All biopsies were forceps